# Aroma Identification and Classification in 18 Kinds of Teas (*Camellia sinensis*) by Sensory Evaluation, HS-SPME-GC-IMS/GC × GC-MS, and Chemometrics

**DOI:** 10.3390/foods12132433

**Published:** 2023-06-21

**Authors:** Yanping Lin, Ying Wang, Yibiao Huang, Huanlu Song, Ping Yang

**Affiliations:** 1College of Tea and Food Science, Wuyi University, Wuyishan 354300, China; 93784063@163.com (Y.L.);; 2Beijing Advanced Innovation Center for Food Nutrition and Human Health, Beijing Research Center for Food Additive Engineering Technology, Laboratory of Molecular Sensory Science, Beijing Technology and Business University (BTBU), Beijing 100048, China; 3Key Laboratory of Agro-Products Processing, Ministry of Agriculture and Rural Affairs, Institute of Food Science and Technology, Chinese Academy of Agricultural Sciences (CAAS), Beijing 100193, China

**Keywords:** tea (*Camellia sinensis*), SOLID phase micro extraction (SPME), two-dimensional comprehensive gas chromatography-olfactometry-mass spectrometry analysis (GC × GC-O-MS), gas chromatography-ion mobility spectrometry (GC-IMS), aroma compounds

## Abstract

Tea (*Camellia sinensis*) is one of the most popular beverages worldwide. Many types of tea products continuously emerge in an endless stream; so, the classification of tea becomes more difficult. Aroma is a vital indicator of tea quality. The present study deals with the identification of aroma compounds in 18 different kinds of tea belonging to three typical tea varieties, including green tea, oolong tea, and black tea, using GC-IMS and GC × GC-O-MS. Moreover, the clustering of all 18 tea samples and the in depth correlation analysis between sensory evaluation and instrumental data were performed using the PCA and OPLS-DA. The results revealed that in all 18 kinds of tea, a total of 85 aroma compounds were detected by GC-IMS, whereas 318 were detected by GC × GC-O-MS. The PCA result revealed that green tea, oolong tea, and black tea could be clearly separated based on their peak areas. The OPLS-DA result showed that a total of 49 aroma compounds with VIP value > 1.0 could be considered as the potential indicators to quickly classify or verify tea types. This study not only compared the aroma differences across different types of teas, but also provided ideas for the rapid monitoring of tea quality and variety.

## 1. Introduction

Tea (*Camellia sinensis*) is the second most consumed beverage worldwide. It has significant pharmaceutical values, such as antioxidant capacity, weight management, and reducing the risk of type 2 diabetes, etc. [1]. Moreover, it also possesses an abundance of flavors that are appealing to the appetites of people [2]. In China, tea can be classified into six types: green tea, white tea, yellow tea, oolong tea, black tea, and dark tea [3,4]. Nowadays, people are increasingly pursuing new tea products with novel and complex aromas; however, green tea, oolong tea, and black tea are still the mainstream tea types all over the world [5]. 

Tea aroma, the soul of tea, is one of the key factors contributing greatly to tea quality and variety [6]. This is mainly because the aroma can reflect the characteristics of tea. Moreover, according to the Chinese standard GB/T23776-1018, “Methodology for sensory evaluation of tea”, launched in 2018, aroma is one of the important indexes for evaluating the tea grade or types with some different and precise descriptive words, such as caramel aroma, fuggy odor, etc. that given by professional sensory evaluators [7]. Even for the same type of tea, there are obvious differences in aromas due to different varieties, processing, and location of growth [8]. 

Those volatile compounds in tea make the aroma come. However, the aroma substances in tea are minimal, accounting for less than 0.1% of its total dry mass [9]. All kinds of aroma compounds in tea have been investigated for many decades. Researchers have been trying to find representative odorants to distinguish between them [2,10,11]. To date, some volatile compounds, such as benzeneacetaldehyde, linalool, linalool oxideII, etc., have been regarded as markers based on their concentrations [10,12]. Feng et al. [13] compared the odorants between six types of teas that were originally processed from one tea cultivar and found that the degradation products of fatty acids were the majority. Baldermann et al. [3] stated that jasmine lactone and indole could be used as indicators to distinguish oolong tea from other kinds of teas based on the peak areas. Zhou et al. [14] developed a new method called the two-step headspace (HS) technique. The research concluded that the determination of volatile compounds could be a dependable manner to distinguish various types of tea or evaluate the tea quality. Nonetheless, substances that can be used as rapid quality/identity indicators during tea production and categorization are still unclear.

To characterize the flavor substances of tea more accurately, a molecular sensory science technique, which is also known as the sensomics approach, has been previously utilized in a tea flavor analysis [12]. It is an effective way to reveal the essential tea aroma at a molecular level by combining mass spectroscopic results (MS) with human olfaction, emphasizing the human-machine integration [15,16]. However, the process in the molecular sensory science technique is somewhat complicated including aroma extraction dilution analysis (AEDA), the calculation of flavor dilution (FD), and odor active value (OAV), aroma recombination and omission experiment [17]. More importantly, when using this approach, an aroma-active compound may not be detectable by MS due to its low threshold and content. For instance, the 2-acetyl-1-pyrroline presenting popcorn perception in oolong tea [18]. Therefore, there are certain limitations associated with this technique, which include rapid classification, quantitative evaluation, identification of off-flavor, and monitoring the processing parameters of tea [19]. 

GC-IMS has been widely used in the past few years due to its outstanding ability in powerful separation and sensitive detection of volatile organic compounds that provide the easy work on food classification and adulteration, the evaluation of food freshness and spoilage, monitoring the processing of food products, and so on [19]. Even though all kinds of tea could be directly and visually distinguished by GC-IMS according to the gallery plot, many compounds failed to be identified mainly because the spectrum library of GC-IMS still needs improvement and supplementation. Hence, different methods should be considered and combined for confirmation and in-depth analysis. GC × GC has been widely applied for a more detailed analysis of volatile compounds in food and proven to be effective due to its high resolution, high sensitivity, and high concentration capacity, which can compensate for the incomplete separation of one-dimensional GC [5,20].

Therefore, to establish a protocol to quickly distinguish between different tea types, in this investigation, 18 kinds of universal and popular tea belonging to three tea varieties, including green tea, oolong tea, and black tea, were selected. Headspace-solid phase microextraction (HS-SPME), which is a simple, highly efficient, and economical aroma extraction method, combined with GC-IMS and GC × GC-O-MS, was utilized to quickly obtain the footprint of all tea samples by identifying their odorants. Most importantly, with the help of professional sensory evaluation and chemometrics, we attempted to find landmark substances that could be used to distinguish easily and quickly between different types of tea. This study aims to provide help or insights into the quality inspection of tea production in the future.

## 2. Materials and Methods

### 2.1. Tea Sample Collection

All 18 tea samples listed in Table 1 were initially grouped and separated into three main types: green tea (No. 1–9), oolong tea (No. 10–14), and black tea (No. 15–18). All tea samples were purchased from Jinglingweng Tea Industry Co., Ltd. (Wuyishan, China) in the same year. All 18 tea samples were the premium teas of three main categories green tea, oolong tea, and black tea. Figure 1 illustrated the geographical locations of all samples. They could basically cover the main tea producing areas in China. All the tea samples were collected and stored at −18 °C to prevent volatile compounds from escaping. Before analysis, the tea samples were ground in liquid nitrogen using a grinder (Baixin Pharmaceutical Machinery Co., Ltd., Wenzhou, China) (3 rounds, 10 s in each round). 

### 2.2. Chemicals

All reference standards (compounds identified by “S” in Appendix A), n-alkanes (C_7_–C_30_), and n-hexane were purchased from Sigma-Aldrich (Shanghai, China).

### 2.3. Sensory Evaluation

Two types of sensory evaluation methods were carried out in this study: one was the professional assessment conducted strictly according to the Chinese standard [21], and the other was called the aroma profile evaluation. The sensory evaluation was carried out blindly and independently by 12 professional panelists, who had a national certificate of tea sensory evaluation. After the discussion by panelists, tea aroma could be separated to 8 attributes, including burning, roasty, fermented aroma, sweet, fruity, floral, fresh, and chestnut-like. A 0–10 point scale with 1 step was applied to evaluate the aroma, of which the higher the value, the stronger the odor intensity. Most importantly, we guaranteed that all samples were edible, non-toxic, and harmless to the human body. We also promised to not disclose any information of the sensory evaluators. All data obtained through this evaluation were used only for the classification of different types of tea on the aspects of sensory studies.

### 2.4. HS-SPME 

SPME is a powerful method for quality control analysis and food differentiation [22]. In this research, 0.5 g of tea powder was weighed into a 20 mL headspace vial, and 5 mL of boiling ultrapure water was added to promote volatile compounds to escape from the tea powder and to simulate the process of tea brewing. Afterwards, the vial was incubated in a thermostatic water bath at 55 °C for 15 min. Extraction fiber (2 cm, 50/30 μm; Supelco, Bellefonte, PA, USA) coated with divinylbenzene/carboxen/polydimethylsiloxane (DVB/CAR/PDMS) materials was then pushed out to extract aromas from the tea. After 40 min of extraction at 55 °C, the SPME fiber was inserted into a GC injector for thermal desorption at 230 °C for 5 min. Each sample was analyzed in triplicate.

### 2.5. Gas Chromatography-Ion Mobility Spectrometer (GC-IMS)

GC-IMS was performed using a GC coupled with an ion mobility spectrometry instrument (Flavourspec^®^-G.A.S. Dortmund Company, Dortmund, Germany), similarly to previously described research [23]. Each 0.5 g tea sample was placed in a 20 mL headspace glass sampling vial, which was then incubated at 55 °C for 15 min at a shaking of 500 rpm. Subsequently, 500 μL of headspace was automatically injected into the injection port on GC-IMS at 85 °C, and the compounds were separated by a WAT-Wax capillary column (30 m × 0.53 mm) at an isothermal condition of 60 °C. Nitrogen (purity ≥ 99.999%) was used as a carrier gas, and the flow rates were varied as follows: 0–2 min (2 mL/min), 2–10 min (2–10 mL/min), and 10–40 min (10–80 mL/min). The draft tube temperature was at 45 °C, and nitrogen was used as a drift gas flowed at a flow rate of 150 mL/min. The data from GC-IMS was analyzed using Laboratory Analytical Viewer (LAV) and GC × IMS Library Search.

### 2.6. Fast Switchable System between GC-O-MS and Two-Dimensional Comprehensive Gas Chromatography-Olfactometry-Mass Spectrometry Analysis (GC × GC-O-MS)

An Agilent 7890B GC-5975B MS (Agilent Technologies, Santa Clara, CA, USA) equipped with a solid-state modulator SSM1800 (J & X Technologies, Shanghai, China) was used, and the instrumental settings were adapted from a previous report with slight modifications [24]. The column temperature was initially set at 40 °C, and after a 4 min holding time, it was increased to 230 °C at a rate of 4 °C/min, at which it was kept for 0.5 min as the final holding time. Three experienced sensory panelists (two females and one male) were recruited to perform a sniffing test on the olfactory output to identifying aroma-active compounds. Ultrapure helium (99.999%, Beijing AP BAIF Gases Industry Co., Ltd., Beijing, China) was used as the carrier gas. Electron-impact mass spectra were generated at ionization energy of 70 eV and an m/z scan range of 50–350. The temperatures of MS ion source and quadrupole were set at 230 °C and 150 °C, respectively.

### 2.7. Statistical Analysis

The GC-IMS data were analyzed using principal component analysis (PCA), and the graphs were plotted using a plugin unit called “Dynamic PCA”, which is embedded analysis software acquired by Laboratory Analytical Viewer (LAV) and GC × IMS Library Search. The stacked bar chart for aroma profile evaluation was drawn using Excel 2021 (Microsoft Office, Redmond, WA, USA) to illustrate the percentage of each sensory attribute in different varieties of tea based on the result of the aroma profile evaluation. The plot of OPLS-DA was analyzed by SIMCA 14.1 (Umetrics, Umea, Sweden). The circle heatmap was developed using TBtools (a Toolkit for Biologists integrating various biological data-handling tools) [25].

## 3. Results and Discussion

### 3.1. Sensory Evaluation

A professional sensory assessment was carried out to determine the grades of the selected 18 tea samples, and the results are presented in Table 1. All 18 tea samples were categorized into grade A due to their comprehensive scores of above 90 according to the China national standard [21]. The colors of tea leaves, liquor colors, and appearances of waste leaves were darker from green tea to black tea; meanwhile, the aroma became intensified due to higher leaf oxidation. Green tea tended to have brisk taste, whereas oolong tea and black tea possessed more sweet taste mainly due to glycoside degradation [26].

According to the aroma profile evaluation results, all 18 kinds of tea could be distinguished based on the eight sensory attributes. The score result was shown in Appendix A. Figure 2 was plotted based on the score results of the aroma profile evaluation. Different colors refer to the eight attributes. Eighteen stripes represent 18 samples. The different lengths of the stripes showed the proportion of a certain sensory attribute in each sample. As shown in Figure 2, chestnut-like and fresh aroma was the feature of green tea (No. 1–9); by contrast, the strengths of the two attributes were much lower in oolong tea (No. 10–14) and were very weak in black tea (No. 15–18). However, the scores of fruity, sweet, and fermented aroma attributes in oolong tea and black tea were higher than those in green tea. This is mainly due to the fact that the oolong tea and black tea were semi-fermented and fermented tea, respectively [27]. In addition, a roast-like aroma attribute was described in all kinds of tea, and this aroma is delivered by heterocyclic compounds formed as a result of the Maillard reaction [2]. The aroma was weak in the green tea because the tea was unfermented. Exceptions were observed in green tea samples no. 7 and no. 8, as these two tea samples originated from the types of roasted green tea [28,29].

### 3.2. Identification of Volatile Compounds by HS-GC-IMS Analysis

GC-IMS was used in this study to initially detect aroma compounds and separate aroma profiles of 18 types of tea.

A total of 85 aroma compounds were detected by GC-IMS, as listed in Appendix A. The volatile flavor fingerprints for all samples are presented in Figure 3. It clearly showed that oolong tea and red tea contained more abundant aroma compounds if compared to the green tea. For the convenience of discussing the results, Figure 3 was divided into six regions (Blocks 1–6) based on the characteristics of the regions where the compounds were located. The result revealed that green tea could be distinctly separated from oolong tea and red tea. This was mainly due to the deficiency of block 3, and this shows that green tea has less sweet and fermented aromas compared to the sensory evaluation results. Meanwhile, volatile compounds that could directly distinguish black tea from oolong tea based on the missing areas of block 1 included (*E*)-2-nonenal, limonene, *β*-pinene, *α*-pinene, and 3-methylpentanol, while those based on block 2 included aroma compounds furfural, 5-methylfurfural, ethyl octanoate, 2-acetylfuran, 2,5-dimethylpyrazine, and 2-methylpyrazine. Terpene and heterocyclic compounds were ignored in previous studies [10,13]. The aroma compounds in block 5 were abundant in all kinds of tea, indicating that they are potent volatile compounds in tea types. Furthermore, almost all types of green tea had same fingerprint, except for the samples No. 1 and No. 3, which was mainly because of the individual areas of block 6 (including odorants ethyl 3-methylbutanoate, 2,3-pentanedione, 2-hexanol, acetic acid, methyl benzoate, 1-hydroxy-2-propanone, and (*E*)-3-hexenol) and block 4 (including odorants acrolein, diallyl disulfide, (*E*,*E*)-2,4-heptadienal, cyclohexanone, octanal, methyl heptanone, (*E*)-2-octenal, and nonanal), respectively.

For the oolong tea, samples no. 10 and no. 14 were significantly different from other samples. The oolong tea samples no. 10 and no. 14 belong to Wuyi Rock tea. This type of tea undergoes a unique roasting process that contributes to a more distinct roast aroma [18]. This is the main reason why the odorants in block 2 and parts of block 3 had higher intensities compared to other oolong teas (the related compounds contained trimethylpyrazine, o-xylene, ethyl 2-methylbutanoate, propyl acetate, ethyl acrylate, ethyl lactate, butyl propionate, propyl acetate, 2-octanone, (*Z*)-3-hexenyl acetate, 1-octen-3-ol, 2-heptanone, 1-propanol, propyl acetate, 2-pentylfuran, 2-butanone, 2,3-butanedione, benzaldehyde, and ethyl propanoate). These odorants could present the two sensory attributes of fruity and roast-like aromas because their scores for both were higher than those of other oolong teas, as shown in Figure 1. On the one hand, the roast-like aroma could be attributed to heterocyclic compounds mainly formed by the Maillard reaction. On the other hand, the fruity aroma was the result of alcohols, ketones, and esters mainly formed by the degradation of lipids or glycosides [2]. The no. 12 sample (Tie guan yin) has been previously reported as being famous for its Yin flavor and distinct orchid fragrance [30]. According to the PCA plot, the odorants (including aroma compounds (*E*)-2-nonenal, limonene, *β*-pinene, *α*-pinene, and 3-methylpentanol) in block 1 might be used as markers to separate no. 12 tea from other oolong teas.

The No.18 sample (Zheng shan xiao zhong) from the black tea group, known for its strong burning aroma notes, was separated from other three black teas. This is mainly because the processing of no. 18 requires a special smoking procedure using pine needles or pine firewood, which differs from the processing of other black teas [31,32]. This is why the color of certain odorants in block 3, including ethyl lactate, butyl propionate, pentyl acetate, 2-octanone, 1-hexanol, 2-butoxyethanol, 2-cyclohexenone, dimethyl trisulfide, and (*E*)-3-hexenol, appears intense in Figure 3.

PCA was conducted in this study to cluster the samples and quickly determine the types of unknown samples. The result in Figure 4 was highly consistent with the above Figure 3 analysis, especially for samples no. 1 and no. 3 of green tea, which were the two exception outlier samples. Thus, the PCA result confirmed that GC-IMS could be used as one of the effective approaches to cluster the samples.

### 3.3. Identification of Aroma-Active Compounds in 18 Kinds of Teas by Two-Dimensional Comprehensive Gas Chromatography-Olfactometry-Mass Spectrometry Analysis (GC × GC-O-MS) Analysis

In this study, over 300 kinds of aroma compounds in all 18 tea samples were identified (Appendix A) using the GC × GC technique. The total ion chromatogram (TIC) plots shown in Figure 5 could directly discriminate green tea from oolong tea and black tea because of the amounts of aroma compounds in samples no. 1–9 were less than those in no. 10–18. In addition, the density of the TIC in no. 6 (Huan shan mao feng) was greater than that in other green teas. This might be because the processing of Huan shan mao feng originally belongs to roasted green tea [33]. Meanwhile, the number of aroma compounds in samples no. 10 and no. 14 was highest among the four oolong teas. This easily explains the importance of the roasting procedure for Wuyi Rock tea [34]. During the fermentation of black tea, more aroma compounds could be generated from the corresponding precursors, such as fatty acids, amino acids, glycosides, etc. [35]. Therefore, the plots of samples no. 15 to no. 18 were also rich and complex.

Although GC × GC possesses strong separation capabilities, only one-third of the 300 aroma compounds could be detected (labeled as “O” in Appendix A) when GC × GC was equipped with an olfactometry system, which was a new approach to carrying out flavor analysis [36]. These volatiles were regarded as aroma-active compounds that could contribute to the aroma profiles of the 18 kinds of tea. These aroma-active compounds exhibited some differences in the aroma strength in each sample, mainly due to variations in their contents. For instance, compound no.121, (*E,E*)-3,5-octadien-2-one could be detected in all 18 samples but could only be identified in oolong tea, black tea, and samples no. 3 and no. 7 of green tea. However, even though compounds no. 7 3-methylbutanal, no. 8 2-methylbutanal, no. 11, 1-penten-3-one; no. 60, 1-octen-3-one; no.89, acetic acid; no. 245, geraniol; and no. 262, 2-phenylethanol, etc. could not be identified in some tea samples, they could be detected in all samples.

### 3.4. Correlation Analysis between Aroma-Active Compounds and Sensory Attributes

Separating undivided compounds using a second column is one of the important advantages of GC × GC [36], but this leads to a very large number of compounds being identified. Aside from that, the olfactometry system is essential and more accurate for aroma analysis [37], but it requires a significant amount of human resources and time. Therefore, it is too difficult to quickly find indicators to distinguish between the varieties of tea samples. Thus, multivariate statistical analysis should be considered. The orthogonal partial least squares-discriminant analysis (OPLS-DA) was employed to analyze the correlation between aroma compounds and sensory attributes (Figure 6). The permutation test shown in Figure 6C demonstrated that the model validation succeeded due to both Q2 and R2 being over 0.5. Figure 6A shows that three kinds of tea were clearly distinguishable. It can be observed from Figure 6B that green tea was more related to the fresh and chest-like aroma, oolong tea was more correlated with the floral notes, and black tea was close to the fruity, sweet, and fermented notes. The Variable Importance for the Projection (VIP) reflects the intensity and explanatory power of the expression pattern of each metabolite that can be used for the classification and discrimination of each sample group, thereby assisting in the screening of marker metabolites (a VIP value > 1.0 is usually used as the screening criterion). VIP values indicate a correlation. The aroma compounds with VIP values > 1.0 were elaborated in Figure 6D, including no.109, linalool; no. 204, nerol; no. 284, trans-nerolidol; no. 148, phenylethanal; no. 245, geraniol; no. 199, methyl salicylate; no. 101, benzaldehyde; no. 318, indole; no. 83, (*Z*)-linalool oxide; no. 262, 2-phenylethanol; no. 78, (*E*,*E*)-2,4-heptadienal; no. 30, *β*-myrcene; no. 50, ocimene; no. 88, (*E*)-linalool oxide; no. 67, 6-methyl-5-hepten-2-one; no. 197, (*E*)-linalool oxide (pyranoid); no. 265, *β*-Lonone; no. 19, hexanal; no. 36, (*E*)-2-hexenal; no. 249, benzyl alcohol; no. 158, dehydrolinalool; no. 35, (+)-limonene; no. 121, (*E*,*E*)-3,5-octadien-2-one; no. 125, *β*-cyclocitral; no. 155, 3-hexenyl hexanoate; no. 188, (*E*)-citral; no. 162, *β*-farnesene; no. 116, 5-methylfurfural; no. 223, nerylacetone; no. 314, creamy lactone; no. 108, 3,5-octadien-2-one; no. 132, hexyl hexanoate; no. 46, 2-pentylfuran; no. 70, 1-hexanol; no. 131, safranal; no. 90, furfural; no. 72, nonanal; no. 15, toluene; no. 91, cis-3-hexenyl butyrate; no. 171, (*Z*)-3,7-dimethylocta-2,6-dienal; no. 139, 1-ethyl-1H-pyrrole-2-carboxaldehyde; no. 200, *α*-farnesene; no. 167, (*E*)-2-hexen-1-yl hexanoate; no. 69, (*Z*)-3-hexenyl-2-methylbutanoate; no. 74, (*Z*)-3-hexen-1-ol; no. 8, 2-methylbutanal; no. 7,3-methylbutanal; no. 87, (*E*)-2-octenal; and no. 66, 2,2,6-trimethyl-cyclohexanone. These aroma compounds were commonly reported in various tea leaves [38,39,40]. Fortunately, most of them could be smelled using olfactometry.

A circular heatmap illustrated in Figure 7 is based on the peak areas of aroma compounds with VIP values > 1.0. It shows that no.109 linalool, which has been widely emphasized in tea aroma compounds [3,10,13], had the highest VIP value and delivered the most floral notes, making it the most important compound among all of the samples. Four black teas showed the highest content of linalool. This is mainly because the thorough fermentation process of black tea promotes the degradation of *β*-primeveroside [2]. However, its content was not always the richest in every type of tea, according to a previous study [41]. This difference in outcome may be the result of different extraction methods. Compound no. 199, methyl salicylate, was another important aroma-active compound found in all of the tea samples. It is a fatty acid derivative and a major contributor to the jasmine-like aroma of tea, especially in black tea, due to full fermentation [39,42]. However, the result of this study showed that it could be considered as one of the indicators to distinguish between different types of teas. Although some aldehydes, such as no.19 hexanal, no. 78 (*E*,*E*)-2,4-heptadienal, and no. 72 nonanal, were also formed from fatty acids, their contents were higher in green tea than in oolong tea and in black tea. This indicates that the fermentation process helps in promoting some complicated reactions, such as lipid degradation, which generates a more complex aroma profile for oolong tea and black tea. However, weak lipid degradation resulted in a fresh aroma note of green tea. The aroma-active compounds no. 265, *β*-ionone and no. 131 safranal were generated from carotenoids [2]. Obviously, carotenoid degradation needs fermentation that can give abundant woody or strong floral aroma to the tea [43]. Moreover, no. 116, 5-methylfurfural and no. 90 furfural were Maillard reaction products [34]. These two aroma compounds were particularly higher in green tea sample no. 1 and no. 10 and oolong tea sample no. 14, which are three different kinds of tea that need roasting during manufacturing [18,44]. In theory, different aromas of different teas are attributed to the different types of tea processing [38], but according to the results of GC-IMS and GC × GC-O-MS, the most direct factor appears to be the degree of fermentation, which was also reported in a previous study [8]. Thus, the function of the shaking process of oolong tea might be similar to that of the fermentation process, which is promoting the degradation of precursors of tea aromas, including carotenoids, lipids, glycosides, and the Maillard reaction. However, different processing methods determine the extent to which different precursor substances participate in the degradation reactions.

## 4. Conclusions

In summary, a total of 85 aroma compounds were detected in 18 kinds of tea by GC-IMS, and 318 aroma compounds were identified by GC × GC-O-MS. The results of GC-IMS were more visible, but those of GC × GC-O-MS were more comprehensive. Moreover, one-third of these were aroma-active compounds that contributed to the overall profile of the tea. Green tea (no. 1–no. 9) contained the least amount of aroma compounds due to the fermentation process, but roasted green tea no.1 had a more abundant aroma profile. Oolong tea (no. 10–no. 14) could be classified into two groups because of the roasting process of no. 10 and no. 14 of Wuyi Rock tea. Black tea (no. 15–no. 18) had the richest aroma profile because carotenoids, lipids, and glycosides in the tea could be fully degraded or oxidized during fermentation. The reason for the different types of tea was mainly attributed to the varieties of aroma compounds. However, the same type of tea can also have different flavors, mainly due to the different content of each odorant. The PCA and OPLS-DA revealed that green tea, oolong tea, and black tea could be clearly separated based on their peak areas. Moreover, the 49 aroma compounds with VIP values > 1.0 could be considered as the potential indicators for classification of the tea types, and these compounds included linalool, nerol, trans-nerolidol, phenylethanal, geraniol, methyl salicylate, benzaldehyde, indole, (*Z*)-linalool oxide, 2-phenylethanol, (*E*,*E*)-2,4-heptadienal,*β*-myrcene, ocimene, (*E*)-linalool oxide, 6-methyl-5-hepten-2-one, (*E*)-linalool oxide (pyranoid), *β*-Lonone, hexanal, (*E*)-2-hexenal, benzyl alcohol, dehydrolinalool, (+)-limonene, (*E*,*E*)-3,5-octadien-2-one,*β*-cyclocitral, 3-hexenyl hexanoate, (*E*)-citral, *β*-farnesene, 5-methylfurfural, nerylacetone, creamy lactone, 3,5-octadien-2-one, hexyl hexanoate, 2-pentylfuran, 1-hexanol, safranal, furfural, nonanal, toluene, cis-3-hexenyl butyrate, (*Z*)-3,7-dimethylocta-2,6-dienal, 1-ethyl-1h-pyrrole-2-carboxaldehyde,*α*-farnesene, (*E*)-2-hexen-1-yl hexanoate, (*Z*)-3-hexenyl-2-methylbutanoate, (*Z*)-3-hexen-1-ol, 2-methylbutanal, 3-methylbutanal, (*E*)-2-octenal, and 2,2,6-trimethyl-cyclohexanone. These aroma compounds are commonly found in various tea leaves. Overall, the identification by GC-IMS and GC × GC-O-MS and the confirmation by the PCA and OPLS-DA could be considered as a method for identifying tea grades or varieties. Therefore, further research applying these preliminary results to actual tea production or classification is also needed.

## Figures and Tables

**Figure 1 foods-12-02433-f001:**
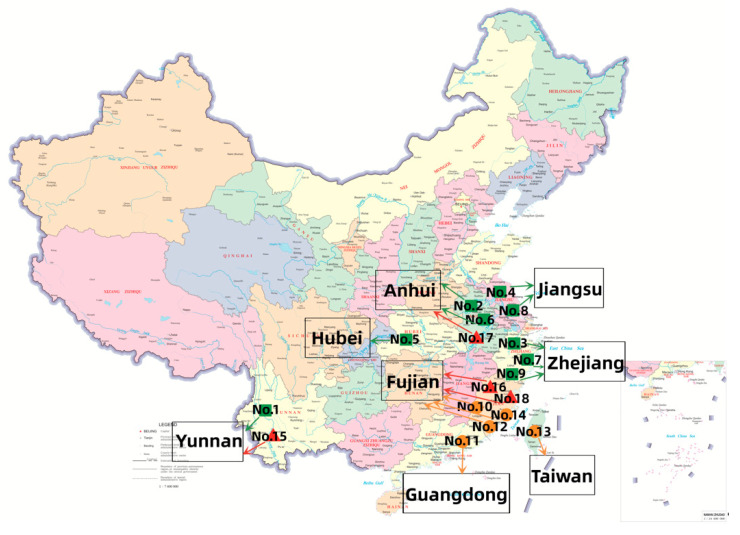
The geographical locations of 18 tea samples.

**Figure 2 foods-12-02433-f002:**
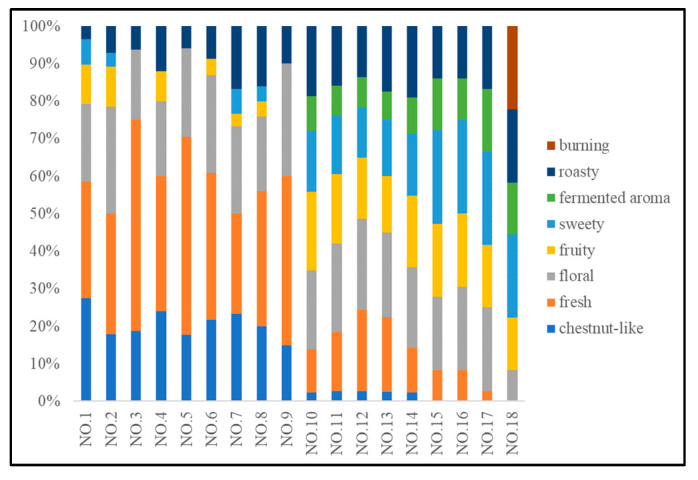
Stacked bar chart for aroma profile evaluation.

**Figure 3 foods-12-02433-f003:**
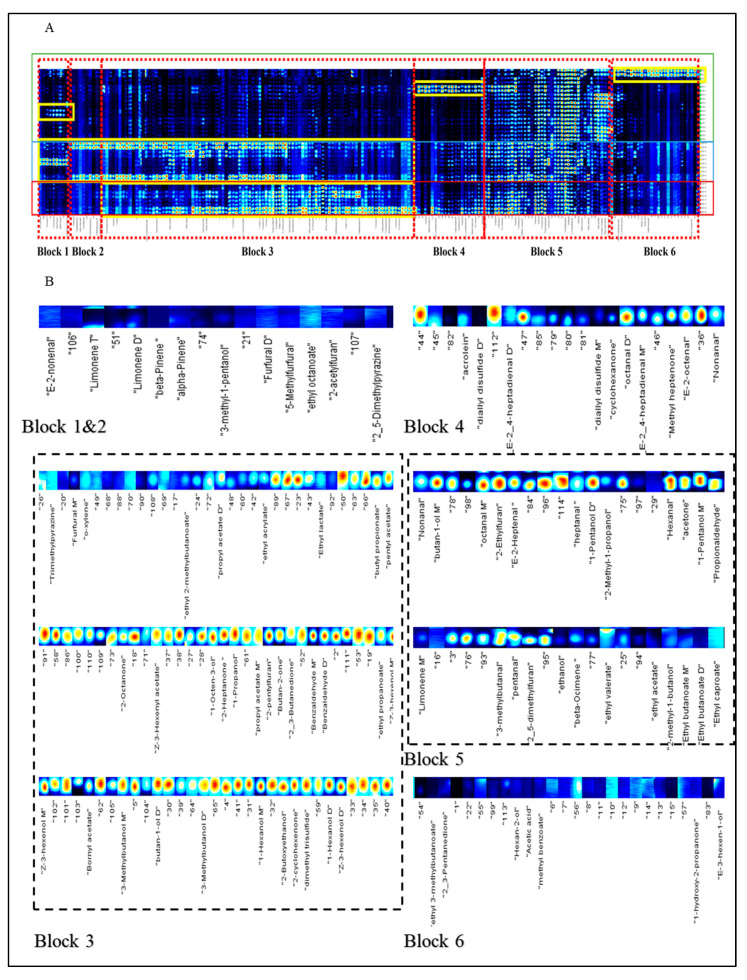
Graphical depiction of the GC-IMS data. (**A**). Gallery plot of the signal peak areas of identified volatile compounds with a visual difference. (**B**). All odorants of different blocks enlarged from plot A.

**Figure 4 foods-12-02433-f004:**
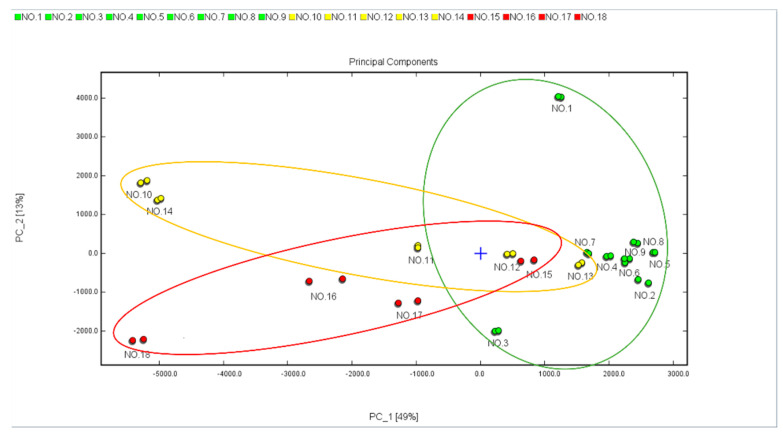
PCA plot based on the results of the HS-SPME-GC-IMS analysis. The color of signal green, yellow, and red colored signals represented green teas, oolong teas, and black teas, respectively.

**Figure 5 foods-12-02433-f005:**
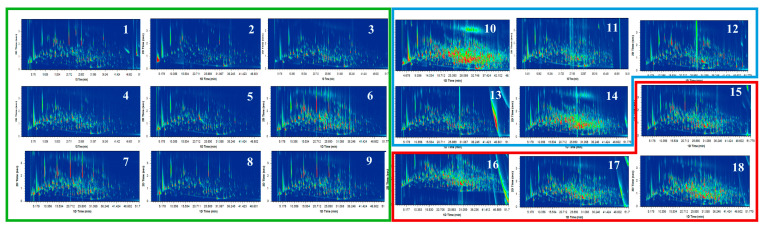
The TIC plots of the 18 tea samples in 2D mode. The areas enclosed by green, blue, and red represent green tea, oolong tea, and black tea, respectively.

**Figure 6 foods-12-02433-f006:**
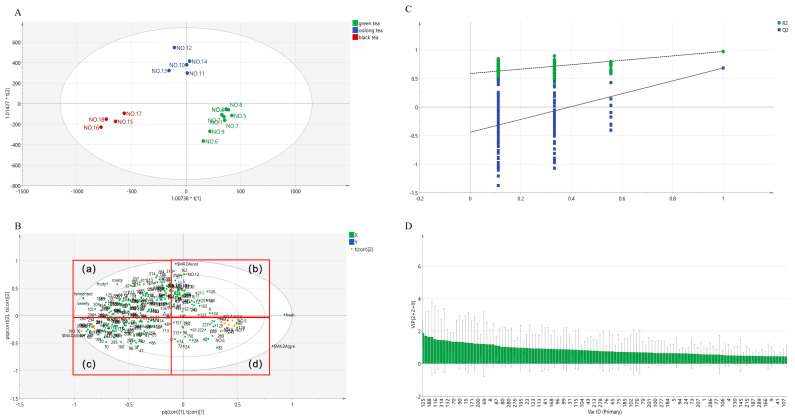
OPLS-DA plots based on the result of the HS-SPME-GC × GC-MS data. (**A**) was the predicted score scatter plot. The green, blue, and red colored signals represented by the green teas, oolong teas, and black teas samples, respectively; (**B**) represented the biplot, with different ellipse areas depicted as 95% confidence regions. The red frame areas (a–d) were enlarged as below respectively; (**C**) represented the validated model; (**D**) represented the variable importance in projection (VIP).

**Figure 7 foods-12-02433-f007:**
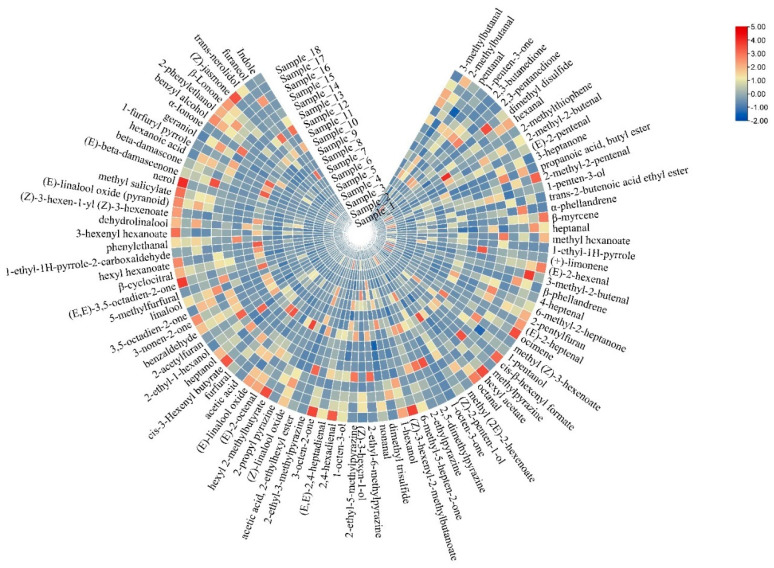
The circle heatmap graph of aroma compounds with VIP >1 in all 18 tea samples.

**Table 1 foods-12-02433-t001:** The professional assessment results of all 18 tea samples.

No.	Tea Samples *	The Shape of a Dry Tea Leaf	Liquor Color	Aroma	Taste	Waste Leaves
Green Teas						
1	Dian qing lu cha	Green, lenitive, uniform, and purity	Green, bright	Refreshing with a tender aroma	Mellow and brisk	Uniform and tender
2	Tai ping hou kui	Flat and heavy, uniform, with no visible pekoe, pale green, veins are dark red	Green, bright	Refreshing with flowery	The brisk and sweet aftertaste	Uniform, tender, and bright
3	Matcha	Tender, slender, uniform, and bright	Green, bright	Refreshing with flowery	Mellow	——
4	Yu hua cha	Needle shape and tender with slight pekoe	Green, bright	Refreshing	The brisk and sweet aftertaste	Uniform, tender, and bright
5	En shi yu lu	Needle shape, dark green, and lenitive with slight pekoe	Green, bright	Refreshing	Mellow and brisk	Uniform, tender, and bright
6	Huang shan mao feng	Heavy, uniform, with pekoe, tender, green	Green, bright	Refreshing	The brisk and sweet aftertaste	Uniform, tender, and bright
7	Long jing cha	Flat with smooth, tight, heavy, tender, green, and lenitive	Green, bright	Refreshing and long-lasting	The mellow and brisk, sweet aftertaste	Uniform, tender, and bright
8	Bi luo chun	Slender, curly, and spiral, with pekoe, silvery green, lenitive, and uniform	Green, bright	Refreshing, with a tender aroma	The mellow and brisk, sweet aftertaste	Tender, uniform, and bright
9	An ji bai cha	Straight, uniform, green, and lenitive	Green, bright	Tender aroma, with long-lasting	The mellow and brisk, sweet aftertaste	White leaf with green vein, uniform
Oolong teas						
10	Da hong pao	Fat, auburn, uniform, and purity	Bright orange-red	Flowery aroma with fruity notes	Mellow and thick, with obviously Yan flavor	Little soft with the toad’s back showing
11	Feng huang dan cong	Tight, auburn, and uniform	Bright orange-red	Flowery aroma	Mellow and thick, obviously Rock flavor	Soft and bright
12	Tie guan yin	Heavy, tender, lenitive, and uniform	Bright yellow-green	Flowery aroma	The mellow and thick, sweet aftertaste	Green leaf with red edge, soft
13	Dong ding wu long	Heavy, dark green, lenitive, and uniform	Green, bright	Flowery aroma	The mellow and thick, sweet aftertaste	Green leaf with red edge, soft
14	Shui xian	Heavy, dark brown, lenitive, uniform, and purity	Bright orange-red	Refreshing, with flowery notes	Mellow and thick, obviously Yan flavor	Green leaf with red edge, soft and bright
Black teas						
15	Jin zhen dian hong	Needle shape and heavy, dark brown with lenitive, gold was revealed	Red, bright	Sweet and aromatic	Mellow and thick	Tender, red leaf, and bright
16	Jin jun mei	Curly, dark brown with gold revealed	Orange and yellow	Flowery aroma with fruity notes, long-lasting	Mellow and thick	Tender, red leaf, and bright
17	Qi men hong cha	Tender, gold was revealed, dark brown, lenitive	Red, bright	Tender, sweet and aromatic	The mellow and brisk, sweet aftertaste	Red leaf, soft, and tender
18	Zheng shan xiao zhong	Tight and lenitive	Orange and red, bright	Cinnamon aroma with caramel notes	Mellow and thick, with an obvious cinnamon flavor	Bronze leaf, tender, a little soft, and uniform

* The overall scores of all 18 tea samples were 90 or above. The evaluation process followed the China national standard [21].

## Data Availability

Data is contained within the article (or Appendix A).

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
