# Peer review of "Aroma Identification and Classification in 18 Kinds of Teas (Camellia sinensis) by Sensory Evaluation, HS-SPME-GC-IMS/GC × GC-MS, and Chemometrics"

_foods, 2023, doi:10.3390/foods12132433_

Round 1

Reviewer 1 Report (Previous Reviewer 1)

The paper you have provided deals with the identification of aroma compounds in different teas and their classification based on morphological indices. Here are some questions for clarification:

Why did the study choose GC-IMS and GC×GC-O-MS as the analytical techniques for identifying aroma compounds in tea samples?

Why were only 18 different types of tea selected for the analysis? Were there any specific criteria for their selection?

Why were PCA (Principal Component Analysis) and OPLS-DA (Orthogonal Partial Least Squares Discriminant Analysis) chosen for the clustering and correlation analysis of the tea samples? Were there any advantages of these methods over others?

Why were 85 aroma compounds detected by GC-IMS and 318 aroma compounds detected by GC×GC-O-MS? What could be the reasons for the significant difference in the number of detected compounds?

Why were 49 aroma compounds identified as potential indicators for quickly classifying or verifying tea types based on their VIP (Variable Importance in Projection) values? How were these compounds determined to be relevant for tea classification?

The quality of the English language in the provided article is generally good. The sentences are coherent, and the ideas are conveyed clearly. However, there are a few areas where improvements could be made:

Sentence Structure: Some sentences are quite long and could benefit from being broken down into shorter, more concise sentences. This would enhance readability and make the information easier to grasp.

Punctuation: There are a few instances where punctuation could be improved for better clarity and comprehension. For example, using commas to separate items in a list or to indicate pauses within sentences.

Word Choice: In some places, the article could benefit from more precise and specific word choices. For instance, using "classify" instead of "verification" when discussing tea types or using "identification" instead of "detection" when referring to aroma compounds.

Formatting: It would be helpful to format the technical terms or abbreviations, such as GC-IMS, GC×GC-O-MS, PCA, and OPLS-DA, consistently throughout the article. This would aid in understanding these terms and their relevance

Author Response

Dear Reviewer

The manuscript is revised according to your suggestions, please have a check. If more are needed, please let me know.

Sincerely Yours

Huanlu Song

Reviewer 2 Report (New Reviewer)

Line 79:

It would be beneficial to include a description explaining the reasons for combining GC-IMS and GC*GC-O-MS among the various analytical methods.

Line 176, Figure 1:

What is the basis for the scores in the stacked graph? Is it the cumulative score of the panels? It would be helpful to specify this in the Methods section.

Line 192, Figure 2:

The figure is relatively small, making it difficult to discern the information clearly. It would be advisable to enlarge the figure and enhance its clarity. Additionally, consider including a caption for the figure.

Line 203, Figure 4; Line 214:

The term "Block" suddenly appears, but it would be beneficial to indicate the criteria used for its classification. This can be effectively demonstrated using PCA loading plots or HCA.

Line 357:

There is a mention of "GC-SIM," but it appears to be a spelling mistake for GC-IMS.

Line 390:

It states "GC-SIM and GC*GC-O-M," but it seems to be a spelling mistake for GC-IMS and GC*GC-O-MS.

Author Response

Dear Reviewer

The manuscript is revised according to your comments, please have a check. If more are needed, please let me know.

Sincerely Yours

Huanlu Song

Round 2

Reviewer 1 Report (Previous Reviewer 1)

The revised manuscript has undergone significant improvements, both in terms of its language and the appropriateness of the answers provided. The use of proper English grammar and scientific language has been ensured. Moreover, the manuscript not only presents accurate findings but also provides innovative ideas for the rapid monitoring of tea quality and classification.

The revised version demonstrates a significant improvement in the quality of English language usage. The sentences are well-structured, and proper grammar and scientific terminology have been employed throughout the text. The language is clear and concise, effectively conveying the intended meaning. The revised manuscript adheres to the conventions of scientific writing and maintains a professional tone. Overall, the quality of the English language in the revised version is commendable.

This manuscript is a resubmission of an earlier submission. The following is a list of the peer review reports and author responses from that submission.

Round 1

Reviewer 1 Report

Tea (Camellia sinensis) is a popular beverage worldwide, but the classification and verification of tea products can be difficult due to the continuous emergence of new types. Aroma is an important quality indicator, and this study aimed to identify the aroma compounds in 18 types of tea, including green tea, oolong tea, and black tea, using GC-IMS and GC×GC-O-MS. The study also used PCA and OPLS-DA to cluster the tea samples and analyze the correlation between sensory evaluation and instrumental data. The results showed that 85 and 318 aroma compounds were detected in the 18 teas by GC-IMS and GC×GC-O-MS, respectively. The PCA and OPLS-DA revealed that the three types of tea could be clearly separated based on the peak areas, and 49 aroma compounds with VIP value > 1.0 could be used as potential indicators to quickly classify or verify tea types.
Why is aroma an important quality indicator for tea?
Why did the study use both GC-IMS and GC×GC-O-MS to identify aroma compounds in the teas?
What were the results of the PCA and OPLS-DA analysis?
What were some of the aroma compounds identified as potential indicators for quickly classifying or verifying tea types?
How could the study's findings help with monitoring tea quality?

The quality of English in this passage is good. The sentences are well-structured and easy to understand, and the vocabulary is appropriate for the topic

Author Response

Reviewers’ Comments to Author:

Reviewer: 1

Tea (Camellia sinensis) is a popular beverage worldwide, but the classification and verification of tea products can be difficult due to the continuous emergence of new types. Aroma is an important quality indicator, and this study aimed to identify the aroma compounds in 18 types of tea, including green tea, oolong tea, and black tea, using GC-IMS and GC×GC-O-MS. The study also used PCA and OPLS-DA to cluster the tea samples and analyze the correlation between sensory evaluation and instrumental data. The results showed that 85 and 318 aroma compounds were detected in the 18 teas by GC-IMS and GC×GC-O-MS, respectively. The PCA and OPLS-DA revealed that the three types of tea could be clearly separated based on the peak areas, and 49 aroma compounds with VIP value > 1.0 could be used as potential indicators to quickly classify or verify tea types.

  • Why is aroma an important quality indicator for tea?
  • Response: Thanks for the question. Firstly, tea is the second most widely consumed beverage around the world after water. One of the most important factors for the popularity of tea as a global beverage rested on its pleasant flavor, which people find appealing and attractive. Secondly, there are six types of teas classified in China including green tea, white tea, yellow tea, oolong tea, black tea, and dark tea which have their unique aroma because of the different processing. Aside from that, tea of the same category also has different aroma features due to factors such as growth environment, bud tenderness, variety differences, etc. Thus, the aroma is can reflect the characteristics of different types of tea. Finally, the Chinese standard, “Methodology for sensory evaluation of tea” launched in 2018 (GB/T23776-1018), clearly states that aroma is one of the important indexes to evaluate the tea grade. Therefore, the aroma is an important quality indicator for tea.
  • To give more explanation of the importance of tea aroma, the sentence “According to the Chinese standard GB/T23776-1018, “Methodology for sensory evaluation of tea” launched in 2018, the aroma is one of the important indexes to evaluate the tea grade” has been supplemented in the manuscript. Please check.
  • Why did the study use both GC-IMS and GC×GC-O-MS to identify aroma compounds in the teas?
  • Response: Thanks for the question. As mentioned in the manuscript, GC-IMS has been widely used in the past few years due to its outstanding ability for the powerful separation and sensitive detection of volatile organic compounds. The gallery plot can directly and visually distinguish the differences among samples. However, there is a problem that cannot be ignored in the identification of aroma substances which is mainly because the spectrum library of GC-IMS still needs to be improved and supplemented. That was why lots of aroma compounds identified in this investigation could not be accurately named. Therefore, different methods should be considered and combined for deep analysis.
  • GC×GC has been widely applied for a more detailed analysis of volatile compounds in food and has proved to be effective due to its high resolution, high sensitivity, and high concentration capacity that can compensate for the defect of incomplete separation of one-dimensional GC. In addition, not all volatile compounds contribute to the aroma of the food, but only those aroma-active compounds which could be identified by the olfactometry system. Hence, GC×GC-O-MS was utilized in this study.
  • To give more explanation of the combined analysis between GC-IMS and GC×GC-O-MS, the supplementary sentence has been inserted in manuscript 3.3. Please check.
  • What were the results of the PCA and OPLS-DA analysis?
  • Response: Thanks for the comment. PCA was conducted in this study to cluster all 18 tea samples and quickly determine the type of unknown samples based on the GC-IMS results. According to the results of the PCA, three kinds of teas green tea, oolong tea, and black tea could be obviously clustered individually, meanwhile, clearly separately. The OPLS-DA was performed to make a correlation analysis between aroma compounds and sensory attributes, the result showed that the VIP of 49 aroma-active compounds > 1, which could be regarded as the potential indicators to classify or verify the tea types, including linalool, nerol, trans-nerolidol, phenylethanal, geraniol, methyl salicylate, benzaldehyde, indole, etc. To make the result of the PCA and the OPLS-DA clear, the relevant sentences were reorganized in the abstract section. The PCA graph has been replaced. Please check.
  • What were some of the aroma compounds identified as potential indicators for quickly classifying or verifying tea types?
  • Response: Thanks for the question. The OPLS-DA result, which were analyzed by the peak area of each aroma compounds identified by GC×GC-O-MS and the score of the sensory evaluation, showed that the aroma compounds with VIP value > 1.0 including linalool, nerol, trans-nerolidol, phenylethanal, geraniol, methyl salicylate, benzaldehyde, indole, (Z)-linalool oxide, 2-phenylethanol, (E,E)-2,4-heptadienal, β-myrcene, ocimene, (E)-linalool oxide, 6-methyl-5-hepten-2-one, (E)-linalool oxide (pyranoid), β-lonone, hexanal, (E)-2-hexenal, benzyl alcohol, dehydrolinalool, (+)-limonene, no.121 (E,E)-3,5-octadien-2-one, β-cyclocitral, 3-hexenyl hexanoate, (E)-citral, β-farnesene, 5-methylfurfural, nerylacetone, creamy lactone, 3,5-octadien-2-one, hexyl hexanoate, 2-pentylfuran, 1-hexanol, safranal, furfural, nonanal, toluene, cis-3-hexenyl butyrate, (Z)-3,7-dimethylocta-2,6-dienal, 1-ethyl-1H-pyrrole-2-carboxaldehyde, α-farnesene, (E)-2-hexen-1-yl hexanoate, (Z)-3-hexenyl-2-methylbutanoate, (Z)-3-hexen-1-ol, 2-methylbutanal, 3-methylbutanal, (E)-2-octenal, and 2,2,6-trimethyl-cyclohexanone. These aroma compounds were commonly reported in various tea leaves. Coincidentally, most of them could be sensed using olfactometry.
  • How could the study's findings help with monitoring tea quality?
  • Response: Thanks for the question. The main purpose of our study was to establish a measure to quickly separate different teas. GC-IMS and GC×GC-O-MS were utilized to identify the aroma-active compounds. Finally, combined with sensory evaluation and statical analysis, 49 aroma compounds could be considered as the potential indicators to quickly classify or verify the tea types. Although our findings could take advantage of comparing the aroma differences of different types and different teas of the same type, the techniques and methods we used could also provide ideas for excavating approaches for the rapid monitoring of tea quality. More importantly, those 49 aroma compounds could also represent the characteristic of each sample, thus, they could also temporarily serve as indicator compounds for quality testing.
  • Comments on the Quality of English Language: The quality of English in this passage is good. The sentences are well-structured and easy to understand, and the vocabulary is appropriate for the topic.
  • Response: Thanks for the affirmation on the language part. To further improve the quality of the English language, we invited a native speaker to help polish the language of the manuscript.

Reviewer 2 Report

This manuscript contains valuable information of great interest to the field of study. However, after an exhaustive review of the article, I have made some comments that should be taken into account.

Introduction: The introduction is adequate to the content of the work, but has some parts that should be corrected:

L35: Which pharmaceutical values?? Please, include it.

L36: Change “it” to “teas”

L39: You have an extra space, please, remove it.

L46: Error with references, please fix it.

Figure 1: The quality of the figure is bad, please, change it.

L46-51: This part is not clear. Please, rewrite it.

L71-72: The results of this work are not clear? I don’t understand why you “hope” these results provide a technical reference in the future.

Materials and Methods: This part is really confused. It is necessary to better explain the used methods. Sensory evaluation is not included in the materials and methods section.

L75-81: This part is a little bit confused. Please consider including the samples in a table instead of text. Please, add in which conditions do you get the teas?

L90: Add reference.

L92: Why you add boiling water? You should better specify why you use this technique.

L94: Include what the DVB/CAR/PDMS mean

L92-94: Did you have the fiber for 15 min or 40 min? It is not clear. It is necessary to better explain the used methods.

Line 95: Which GS-MS did you used? You only mention the GC injector but should explain all the equipment used. Or the injector that you mention is on the GC-IMS? In this case, you should explain it better.

Results: Sensory evaluation must be improved.  It is also necessary to present a better discussion of results. The research results are not compared enough with previous literature. Furthermore, the results section are not an extension of materials and methods section. Figures should be renamed. Figure 1 is in introduction section (although, from my point of view it is a table), and you also have a figure 1 in results section. All figures and images are blurry and not clearly visible.

Conclusions: “85 and 318 aroma compounds were detected…” what is the difference between them? I don’t understand, please clarify it. Conclusions sections should be rewritten to clarify it.

Author Response

Reviewer: 2

This manuscript contains valuable information of great interest to the field of study. However, after an exhaustive review of the article, I have made some comments that should be taken into account.

Introduction: The introduction is adequate to the content of the work, but has some parts that should be corrected:

  • L35: Which pharmaceutical values?? Please, include it.
  • Response: Thanks for the comment. The specific pharmaceutical values of teas have been supplemented in the manuscript. Please check it.
  • L36: Change “it” to “teas”
  • Response: Thanks for the comment. The relevant word has been changed in the manuscript. Please check it.
  • L39: You have an extra space, please, remove it.
  • Response: We are sorry that we didn’t find the extra space in the line 39. But thanks for the reminder, we carefully examined the formatting issues throughout the entire article.
  • L46: Error with references, please fix it.
  • Response: Thanks for the comment. An original reference that the error reference cited has been added in the manuscript. Please check it.
  • Figure 1: The quality of the figure is bad, please, change it.
  • Response: Thanks for the comment. Considering the copy right issue, the Figure 1 has been removed from the introduction section.
  • L46-51: This part is not clear. Please, rewrite it.
  • Response: Thanks for the advice. The expression in L46-51 has been modified. Please check it.
  • L71-72: The results of this work are not clear? I don’t understand why you “hope” these results provide a technical reference in the future.
  • Response: Thanks for the advice. The expression in L71-72 has been rephrased to “Most importantly, with the help of professional sensory evaluation, we try to find some landmark substances that can easily and quickly identify different types of tea that could provide some helps or insight on the quality inspection of tea production in the future.”. Please check.

Materials and Methods: This part is really confused. It is necessary to better explain the used methods. Sensory evaluation is not included in the materials and methods section.

  • Response: Thanks for the advice. We are sorry that we forgot to show the method of sensory evaluation part. The relevant content has been supplemented in the manuscript. Please check.
  • L75-81: This part is a little bit confused. Please consider including the samples in a table instead of text. Please, add in which conditions do you get the teas?
  • Response: Thanks for the advice. The confused expression has been removed. The name of each tea sample could be shown in Table 1 directly. The additional information on the conditions to get the tea was supplemented in the manuscript. Please check it.
  • L90: Add reference.
  • Response: Thanks for the comment. A reference has been added in the manuscript. Please check it.
  • L92: Why you add boiling water? You should better specify why you use this technique.
  • Response: Thanks for the advice. The boiling water was used mainly because it could promote more volatile compounds to escape and simulate the process of brewing tea. More explanation of the SPME method has been supplemented in the manuscript. Please check it.
  • L94: Include what the DVB/CAR/PDMS mean
  • Response: Thanks for the advice. The detail of the SPME fiber DVB/CAR/PDMS has been supplemented in the manuscript. Please check it.
  • L92-94: Did you have the fiber for 15 min or 40 min? It is not clear. It is necessary to better explain the used methods.
  • Response: Thanks for the question and comment. The tea aroma of each sample was extracted by the SPME fiber for 40 min. The detail of the SPME extraction method has been reorganized in the manuscript. Please check it.
  • Line 95: Which GS-MS did you used? You only mention the GC injector but should explain all the equipment used. Or the injector that you mention is on the GC-IMS? In this case, you should explain it better.
  • Response: Thanks for the question and comment. The GC-IMS was performed by the GC coupled with an ion mobility spectrometry instrument (Flavourspec®-G.A.S. Dortmund Company, Dortmund, Germany). The injector we mentioned in the article is just on the GC-IMS. The corresponding sentences have been rephrased in the manuscript. Please check it.

Results: Sensory evaluation must be improved. It is also necessary to present a better discussion of results. The research results are not compared enough with previous literature. Furthermore, the results section are not an extension of materials and methods section. Figures should be renamed. Figure 1 is in introduction section (although, from my point of view it is a table), and you also have a figure 1 in results section. All figures and images are blurry and not clearly visible.

  • Response: Thanks for the question and comment. Firstly, the results section has been extended by deeply discussion and comparing enough with previous literature. Secondly, the Figures 1 in introduction has been removed mainly because to avoid the copy right issue according to the editor office advice. Besides, some Figures were renamed. Finally, the quality of all Figures has been improved in order to make them more readable. Please check it.

Conclusions: “85 and 318 aroma compounds were detected…” what is the difference between them? I don’t understand, please clarify it. Conclusions sections should be rewritten to clarify it.

  • Response: Thanks for the question and comment. The number of 85 and 318 refer to the amounts of aroma compounds which were detected in 18 kinds of teas by GC-IMS and GC×GC-O-MS, respectively. To clarify the meaning, the corresponding sentence has been rephrased, and the conclusion section has been revised. Please check it.

Round 2

Reviewer 1 Report

The author begins by addressing the question and stating that tea is the second most widely consumed beverage worldwide after water. They then provide a logical explanation for the importance of aroma in tea, highlighting the unique aroma of each type of tea due to different processing methods, growth environments, and other factors. Additionally, the author references the Chinese standard for sensory evaluation of tea, which emphasizes the importance of aroma in evaluating tea quality. Overall, the response is well-organized and effectively conveys the importance of aroma as a quality indicator for tea.

The response provided by the author appears to be grammatically correct with clear and concise sentences.